# What Is Social about Autism? The Role of Allostasis-Driven Learning

**DOI:** 10.3390/brainsci11101269

**Published:** 2021-09-25

**Authors:** Meshi Djerassi, Shachar Ophir, Shir Atzil

**Affiliations:** Department of Psychology, Hebrew University of Jerusalem, Jerusalem 9190501, Israel; meshi.djerassi@mail.huji.ac.il (M.D.); shachar.ophir@mail.huji.ac.il (S.O.)

**Keywords:** autism, learning, allostasis, social development, domain-general neural circuits, multi-modal integration, parent-infant synchrony

## Abstract

Scientific research on neuro-cognitive mechanisms of autism often focuses on circuits that support social functioning. However, autism is a heterogeneous developmental variation in multiple domains, including social communication, but also language, cognition, and sensory-motor control. This suggests that the underlying mechanisms of autism share a domain-general foundation that impacts all of these processes. In this Perspective Review, we propose that autism is not a social deficit that results from an atypical “social brain”. Instead, typical social development relies on learning. In social animals, infants depend on their caregivers for survival, which makes social information vitally salient. The infant must learn to socially interact in order to survive and develop, and the most prominent learning in early life is crafted by social interactions. Therefore, the most prominent outcome of a learning variation is atypical social development. To support the hypothesis that autism results from a variation in learning, we first review evidence from neuroscience and developmental science, demonstrating that typical social development depends on two domain-general processes that determine learning: (a) motivation, guided by allostatic regulation of the internal milieu; and (b) multi-modal associations, determined by the statistical regularities of the external milieu. These two processes are basic ingredients of typical development because they determine allostasis-driven learning of the social environment. We then review evidence showing that allostasis and learning are affected among individuals with autism, both neurally and behaviorally. We conclude by proposing a novel domain-general framework that emphasizes allostasis-driven learning as a key process underlying autism. Guided by allostasis, humans learn to become social, therefore, the atypical social profile seen in autism can reflect a domain-general variation in allostasis-driven learning. This domain-general view raises novel research questions in both basic and clinical research and points to targets for clinical intervention that can lower the age of diagnosis and improve the well-being of individuals with autism.

## 1. Introduction

Autism is a heterogeneous developmental variation, defined by the American Psychiatric Association as a persistent deficit in social communication and interaction [1]. As such, scientific investigations on autism often focus on social behaviors and their underlying neural mechanisms. While this approach provides insight into autism, it has two major challenges. First, social variation is not an inclusive description of autism, as individuals with autism also exhibit atypical verbal, cognitive, and sensory-motor profiles [2,3,4,5]. Second, the field of social neuroscience still debates the idea that social processing in the brain is produced by a dedicated tissue, or a “social brain” [6,7]. An alternative approach considers social processing and social cognition as a complex neural computation, supported by multiple domain-general processes, including motivation, perception, and motor control [8]. Considering the complex autism phenotype, along with the challenges of neurally defining social processing as a discrete module, we propose that the developmental variation seen in autism stems from variations in domain-general processes, which are not necessarily “social” per se. 

In this Perspective Review, we propose a hypothesis, by which autism is not a “social” deficit, but rather a variation in learning. Specifically, we propose that in social animals, newborns depend on their caregiver for allostasis regulation, which is defined as the ongoing process of regulating the internal milieu [9,10]. This makes the social environment vitally salient and reinforces *allostasis-driven learning* of social knowledge and skills. Given that social knowledge and skills are acquired through learning, domain-general neural processes that underlie learning can impact social development, including perception, motor control, allostasis regulation, and multi-modal integration. Variation in such domain-general processes can affect allostasis-driven learning of social knowledge and skills and manifest as autism. 

To support this view, we review lines of evidence. The first shows that social processing in the brain does not depend on a dedicated neural tissue, but rather involves domain-general neural circuits controlling allostasis-regulation and multi-modal integration. The second line of evidence shows that typical social development depends on allostasis-driven learning. We then review evidence demonstrating that the same domain-general processes of allostasis and learning vary in autism, both behaviorally and neurally. Finally, we integrate the evidence to propose that autism is not the result of a “broken social brain” but the manifestation of a variation in allostasis-driven learning.

## 2. Current Theories on Cognitive Mechanisms of Autism Focus on Social Deficits 

Several theoretical accounts in the literature portray the cognitive and psychological mechanisms that underlie autism [11]. Autism is often explained by impairments in social cognition, including the ability for the theory of mind and empathy. The theory of mind account focuses on the ability to mentalize, which is the ability to internally represent the mind of others, including abstract and hidden information about others’ emotions, intentions, knowledge, and beliefs [12,13]. Individuals with autism have difficulties in mentalizing the mental states of others [14,15], have difficulties inferring others’ emotions and intentions by looking at their eyes [16], distinguishing their own knowledge from knowledge attributed to others [17,18], and attributing mental states spontaneously [19]. Moreover, the individual ability for the theory of mind is associated with the severity of further autistic symptoms, such as impeded social communication and repetitive restrictive behaviors [20]. According to the Double Empathy Problem account [21], individuals with autism are not only impaired in understanding others but are also less understood by neurotypical individuals. At the level of the brain, regions that are implicated to support the theory of mind and empathy display atypical activation and connectivity patterns among individuals with autism. Specifically, the medial prefrontal cortex (mPFC), and the posterior and anterior cingulate cortices (PCC and ACC, respectively), which are often referred to as the default mode network and are associated with theory of mind performance [22,23], have decreased connectivity in individuals with autism [24,25,26], as well as atypical structure and activations pattern [27,28,29,30]. Moreover, individuals with autism have an atypical developmental trajectory [31] and impaired functional connectivity [25,32] between the temporo-parietal junction and the inferior frontal gyrus, which were also associated with the theory of mind [33,34].

In addition to social cognition, other accounts explain autism by decreased internal motivation for social interactions [11]. For example, the Social Motivation theory posits that individuals with autism experience less internal rewards from social interactions compared to neurotypical individuals, and seek less social contact [35,36]. This is often explained in terms of reduced “salience” or the importance of social information for individuals with autism [35]. In line with this, the Social Orientating hypothesis [37] predicts the earliest indicator of autism is the lack of attention to social stimuli. Neuroimaging research shows that individuals with autism have atypical function in the orbitofrontal–striatum–amygdala circuit and expression of oxytocin receptors within it, which are implicated in reward and motivation [35,38,39,40]. This finding is highly consistent across the neuroimaging literature in the field of autism research [41].

Both the social cognition and social motivation accounts capture important variations seen in autism, each focusing on a distorted function in a specific social feature, and its underlying neural circuitry. Yet, can social motivation and social cognition be considered dedicated modules that are categorically separated from non-social motivational and cognitive processes? Other accounts explain autism using more general cognitive processes. For example, the Executive Function theory [42,43], suggests that autism is caused by impairments in the ability to organize actions and thoughts, and control impulses. Other theories propose that the core deficit in autism is differentiating important information from the context, which makes it difficult for individuals with autism to process complex stimuli and shift attention. These accounts include the Weak Central Coherence theory [44], the Context Blindness theory [45], and the Monotropism theory [46]. According to the Extreme Male Brain theory [47], autism reflects an extreme form of the typical male profile, favoring systemizing over empathizing [47,48]. Individuals with autism are superior in tasks that favor systemizing [49,50], have increased attention to detail [50], and are impaired in tasks of empathy and theory of mind [16,19,51]. This account is supported by the significantly higher prevalence of autism in males compared to females [52]. Moreover, males and females with autism are more similar to each other than to neurotypical males or females, respectively, in neural connectivity [53] and grey matter volume in the left inferior parietal lobe and operculum [54]. It is suggested that elevated exposure to testosterone during pregnancy could impact the neural development of the fetus, favoring extreme masculinization of the brain [55].

We propose that not only autism, but sociality at large is not a dedicated module and that its mechanisms can be explained with a domain-general account. Specifically, social processing does not depend on a dedicated neural system. Instead, social processing involves domain-general neural mechanisms that control allostasis-regulation and multi-modal integration. These circuits support the processing of both social and non-social information, with no categorical distinction between “types” of information [8]. The idea that social processing is not a dedicated module but rather, is supported by domain-general neural circuits, has implications for the conceptualization of autism and its underlying neural mechanisms.

## 3. Neural Basis of Social Processing and How It Varies in Autism

A large body of evidence from the field of Social Neuroscience portrays the neural circuits that are consistently involved in social processing. Specifically, neural activity and connectivity within a group of cortico-limbic brain regions are consistently involved in the neural processing of social information. These regions include the subcortical regions in the amygdala, nucleus accumbens (NAcc), and hypothalamus [56,57,58,59], and cortical regions in perception and action cortices [60], limbic cortices in the ventral anterior insula and cingulate cortex [61,62], and other association cortices in the ventromedial prefrontal cortex (vmPFC), ACC, and PCC [63,64]. Moreover, coordinated function and dopamine secretion within these regions is associated with improved social behavior in humans [65,66,67] and non-human mammals [68,69,70]. 

While these regions are repeatedly reported to underlie social processing across the literature, they are not dedicated or specific to social processing. The amygdala, NAcc insula, and ACC have a visceromotor role and via the hypothalamus, they regulate the internal milieu of the body through the autonomic nervous system [71], and the endocrine system [71,72,73]. The amygdala, dorsal ACC, and ventral anterior insula are also considered part of the salience network, which engages when facing important information in the world [74,75,76], and together with sensory and motor regions patriciate in perception and action in salient situations, whether social or non-social [77,78,79,80]. Association cortices in the vmPFC and PCC are involved in the representation of information about the world, including rudimentary and abstract concepts, and are considered to be part of the default mode network, which has a role in mentalization and theory of mind [22,23]. Altogether, the evidence suggests that social processing in the brain relies on domain-general processes of visceromotor control on the internal milieu of the body (allostasis), perception-action in face of salient stimuli, and representation of abstract knowledge (Figure 1).

Individuals with autism show atypical connectivity within the salience and the default-mode networks [82,83,84]. Children with autism show hyper-connectivity within regions of the default-mode network, compared to neurotypical children, which is associated with the degree of their social impairment [24,26]. Adults and adolescents with autism show reduced connectivity in the default mode network compared to neurotypical controls [85], and children with autism display hypo-connectivity in the salience network, compared to neurotypical children, which is associated with sensory and social symptoms [86]. In addition to the default mode and salience networks, altered connectivity patterns are also found in other domain-general neural circuits among individuals with autism. This includes weaker connectivity between the amygdala and the medial prefrontal cortex [38,87], between both the amygdala and the PCC and temporal lobes [87,88], and between the amygdala and the striatum [87]. Atypical function in this circuit is associated with decreased social motivation and social saliency [35]. Altogether, when considering the neural phenotype of autism in humans, the domain-general nature of the neural circuits that are affected in autism supports the idea that the neural variation that underlies autism is not exclusive to social processing.

## 4. Social Development Depends on Allostasis-Driven Learning 

### 4.1. Allostasis Regulation Shapes Social Learning 

Optimal social development and well-being in children depends on the extent to which infants and parents are attentive to one another, and reciprocate affective states and behavioral communication cues, a term coined as parent-infant bio-behavioral synchrony [89,90]. The idea that synchronous parental care is adaptive for children is intuitive. However, the mechanisms in the child via which synchrony promotes learning and development remain unknown. In order to understand the mechanisms of social development in children, a key scientific challenge is to recognize key processes in the child via which parent–child synchrony in early life optimizes learning and development.

Recent literature points out that parent–child synchrony is adaptive because it helps to regulate the biological and psychological demands of the child, or their allostasis [8]. Since human infants are helpless in maintaining their ongoing regulatory needs, they depend on a dedicated caregiver for allostasis regulation. Parents feed their infants to regulate their diet and immune system, sing and touch their infants to regulate their arousal and temperature, and control many aspects of the infants’ autonomous nervous system [8,91,92]. Parent–child synchrony is a useful strategy for the social regulation of allostasis, which is highly rewarding to both parents and children, and effectively reinforces physical well-being, as well as social bonding and social learning. By being cared for, infants learn to socially interact in order to meet their allostatic demands, as biological primal needs for survival motivate infants toward social learning. Therefore, it is not synchrony per se that supports development and learning but rather its regulatory consequences on children’s allostasis [8,93].

As allostasis regulation is key for social development and learning [8,94,95], variation in children’s ability for allostasis regulation can adversely affect the rewarding experience of social interactions. This can result in reduced social orienting, social seeking, social liking, and as a result, deficient social learning, which are central characteristics of autism [35]. Indeed, children with autism show impaired patterns of allostasis regulation (see a review of empirical evidence in the next section), and it is suggested here that this can shift the trajectory of learning and by that determine the developmental phenotype of autism.

### 4.2. The Social Environment Shapes the Multi-Modal Representations of Perception and Action Patterns

The human brain is relatively immature at birth. One of the prominent neural features that are missing in newborns is the ability for multi-sensory integration, whereby information from different modalities is synthesized and used in concert [96]. This integrative capability gradually develops during postnatal life as the brain gains experience perceiving consistent multi-modal patterns in the environment, and gradually representing them as rudimentary concepts [8,97,98]. Since humans are social animals and infants completely rely on the caregiver for survival, the infant’s environment is essentially social, and the most prominent sensory patterns in the infant’s environment are of other humans. As a result, the first rudimentary concepts that infants acquire are social concepts [8,92]. For example, by repeated interaction with the caregiver, infants gradually recognize the spatial organization of the caregiver’s visual, auditory, olfactory, and tactile features and form a concept of *mommy* (or any other *caregiver*) [8].

*Mommy* is not only the most consistent pattern of information in the environment, it is also the most salient and potentially rewarding because it is essential for the infant’s allostasis. By integrating exteroceptive social information about the caregiver with interoceptive rewarding information about allostasis, the *caregiver* concept is quickly learned and repeatedly reinforced. High statistical regularity in the integration between the *caregiver* and *allostasis regulation* promotes social motivation and reinforces learning of social concepts and behaviors. As caregivers regulate infants’ allostasis, infants gain experience not only in rudimentary perceptual concepts, but also in more abstract concepts such as emotion concepts, and representations of others’ minds [8,95], and perception-action concepts, such as *breastfeeding*, *synchrony*, or *vocalizations* [95]. Thereafter, infants learn to share their attention with the caregiver [99] and to coordinate explicit knowledge (social and non-social) by acquiring language [100]. Synchronous parenting fosters the child’s ability to learn and use concepts [101] and can determine the content of concept learning by infants. For example, parental use of mental state language during parent–infant interactions promotes children to label their own emotions [102] and later, to infer and represent other people’s mental states [95,102,103,104].

Learning concepts and skills is a key domain-general process that underlies typical development. Social regulation of allostasis during development promotes social motivation and provides a strong driving force for social learning. However, social learning is not a special type of learning but rather more pronounced in early life. This is because of the high relevance of social information to the infant’s allostasis and the high prevalence of social information in the infant’s environment (see Box 1 and Figure 2).

Box 1Learning in Typical Social Development and Autism.1. ConceptsConcepts (C) are multi-modal representations of statistically regular patterns of information in the external world and the internal milieu. A central aspect of development is constructing concepts. During development, infants detect statistical regularities in the environment and organize them into concepts [105,106,107]. Concepts can be thought of as spatial and temporal patterns of information that enable the brain to perceive the world in a multi-modal meaningful way and to prepare for upcoming allostatic and environmental demands [71,108]. Since humans are social animals and depend on their caregiver, the first concepts infants acquire are social [8]. These include rudimentary sensory concepts such as a “face” to more abstract concepts like “mommy”, and more complex sensory-motor concepts like “breastfeeding” and “synchrony” or “crying” [95]. Consistent patterns of non-social exteroceptive-interoceptive-motor information are similarly constructed as concepts, and there is no categorical difference between social and non-social learning.
C(t)=f(introception(t),exterocepton(t), motor(t))C is a function of mental representations of consistent multi-modal patterns of information from interoceptive, exteroceptive, and motor sources (Figure 2).
C(t+1)=C(t)+η(t)⋅ΔILearning and updating of concepts depends on the rate of learning (η) novel information (ΔI) from interoceptive, exteroceptive, and motor sources.2. Learning Concepts Depends on AllostasisThe elementary ongoing process of optimizing the body’s internal milieu shapes learning through motivation and reward. Patterns in the world that are relevant for allostasis (learning them reinforces allostasis regulation) are efficiently learned as concepts.The rate of learning (η) of a concept depends on its impact on allostatic regulation

$$\eta \left(t\right)\propto \frac{\partial \log\left(A\right)}{\partial \log\left(C\right)}$$

Here, the sensitivity of allostasis to a certain concept is represented as (∂log(*A*))/(∂log(*C*)). For example, concepts (*C*) with a large impact on allostasis (*A*), including social concepts such as *mommy* or any other caregiver, are learned faster than concepts with minor impact on allostasis, such as many non-social concepts.3. AutismAccording to this model, autism results from an atypical rate of learning, due to variation in domain-general processes of perception (interoception, exteroception), motor control, allostasis regulation, or their multi-modal integration into concepts. Variation in any of these processes can interfere with the rate and proposition of concepts and manifest as atypical social development. These domain-general processes are not orthogonal to one another, but rather represent different levels of observation, which can be prognostic of autism. For example, an atypical phenotype in infants in physiological measurements of allostasis, behavioral measurements such as synchrony with the caregiver, or neural measurements such as the formation of multi-modal neural tracts can serve as bio-behavioral indicators for future diagnosis of autism.

## 5. Individuals with Autism Show Variations in Domain-General Processes of Learning

The hypothesis that autism results from a variation in allostasis-driven learning raises specific predictions by which individuals with autism show an atypical phenotype in the domain-general processes of perception (interoception, exteroception), motor control, allostasis regulation, or their multi-modal integration. In the next section, we review evidence that the basic ingredients of allostasis-driven learning vary in autism.

### 5.1. Individuals with Autism Show Atypical Patterns of Allostasis Regulation

Empirical studies demonstrate that individuals with autism have atypical patterns of allostasis regulation [112,113]. Specifically, children and adults with autism exhibit atypical eating patterns [114], excessive water drinking [115], and disturbed sleeping patterns [116]. Such disturbed control over allostatic processes potentially stems from disturbances in the accurate perception of the internal milieu or interoception. Individuals with autism have reduced awareness of interoceptive signals of their body, such as reduced thirst awareness [117], reduced performance in heartbeat tracking tasks [118], and hyposensitivity to pain and proprioception [119], compared to neurotypical individuals. An infant’s ability for interoception is necessary to ensure attuned parental care, which is adapted to the infant’s allostatic needs. For example, to be fed, the infant’s brain must sense a decrease in plasma glucose levels through interoception, and socially communicate it to the caregiver via motor control over cry. This signals the allostatic need to the caregivers and elicits parental response aimed to regulate the infant allostasis. By attending to the infant’s allostatic needs, the parent reinforces further social behavior in the infant [8]. Disrupted interoception in infancy can interfere in three central processes that impact social development. First, it impairs the infant’s signals for regulation and therefore the caregiver’s ability to accurately regulate the infant, and later-on the infant’s ability for self-regulation [120,121,122]. Second, it impairs social motivation of the infant by reducing the rewarding value of social care. Third, it impairs social learning by disrupting the statistical regularity of allostatic-based reinforcement derived from social care (see Box 1).

### 5.2. Individuals with Autism Show Atypical Perception and Motor Function

In addition to altered interoception in autism, several studies report perceptual variation in autism across different exteroceptive modalities [123,124,125,126,127]. For example, individuals with autism have altered visual perception, including superior performance in detail or pattern recognition tasks [128,129], and in visual search tasks [130], along with significant deficits in motion processing tasks [131,132]. Auditory perception variation in autism is also reported [133,134], showing that individuals with autism tend to have improved pitch memory [135] but lower recognition of speech in noise [136] compared to neurotypical controls. Individuals with autism also show a variation in motor function [137] and often show stereotypical repetitive motor movements [138] and stereotypical vocalizations [139]. A growing body of evidence highlights the importance of such sensory-motor variations in the prognosis of individuals with autism [140]. Importantly, the sensory phenotype of children with and without autism significantly correlates with their social skills [141,142]. Thus, sensory-motor variations can impair learning (both social and non-social), as it can impair the formation of multi-modal associations and constructing concepts. Given the social environment human infants develop in and the importance of social interactions to the infant’s allostasis, many perception–action patterns that infants learn are social. For example, forming action-dependent concepts and their quick updating is crucial for synchronization and communication with the caregiver during social interaction. Evidence shows that motor control over vocalizations is crucial for synchronous social behavior [143,144], and mother–infant attunement [67]. As such, variation in perception and in motor functions during development can result in atypical trajectory of social development. 

### 5.3. Individuals with Autism Show Atypical Learning

Impairments in learning among individuals with autism are reported at the neural and behavioral levels. At the neural level, evidence in multiple animal models of autism shows that genetic mutations associated with autism cause altered expression and translation of dendritic proteins that mediate synaptic plasticity [145,146,147,148,149,150]. In humans, individuals with autism display impairments in long-term potentiation evoked by transcranial magnetic stimulation [151,152]. At the behavioral level, individuals with autism show impairments in reinforcement learning [153,154,155]. Reinforcement learning is guided by the motivational state of the organism approaching a reward or avoiding a punishment [156,157]. Allostatic demands impact the motivational state of an individual and are therefore a central organizing principle of the nervous system, which guides learning and behavior [158]. Dopamine transmission in cortico-striatal circuits is central in learning about rewards and punishments and disturbances of the dopaminergic system in cortico-striatal circuits have a key role in psychopathologies that involve reinforcement learning [158,159,160]. Individuals with autism do not conform to standard patterns of reinforcement learning and are less affected by rewards and punishments [153,155,161]. Accordingly, autism is associated with aberrant patterns of dopamine signaling in the ventral tegmental area and substantia nigra, affecting reward processing and goal-directed behavior [162]. The distorted patterns of reinforcement-learning support the significance of allostasis-driven learning in autism. In addition to reinforcement learning, individuals with autism show impaired motor learning, specifically in tasks involving the acquisition of novel movement patterns acquisition [163], and implicit procedural learning tasks [164]. Moreover, individuals with autism show impaired language learning, and up to a third of children with autism are minimally verbal [165]. Among verbal children with autism, language is acquired in a substantial developmental delay. While typically developing children produce their first words at the ages of 8 to 14 months, the mean age of first word-use in children with autism is three years [166]. Importantly, the age of first word-use is correlated with the prognosis of children with autism [167]. Individuals with autism also display an atypical use of language, for example, children with autism over-imitate phrases they hear (“echolalia”, [168]), and exhibit syntactic impairments [169]. Such impairments in language acquisition can reflect impairments in concept use and acquisition. Children and adults with autism show differences in conceptualizations and have difficulties categorizing new information by forming prototypes and generalizing previously learned concepts to new situations [170]. Leider and colleagues recently demonstrated that individuals with autism rely more on longer-term statistics rather than recent events and show a slower motor response to changing sensory patterns of non-social stimuli. This suggests that individuals with autism are slower to form and represent statistically regular patterns of information [171] (i.e., concepts), possibly because of aberrant updating of probabilistic representations of the environment [172]. This evidence emphasizes that the different aspects of the autism phenotype, from low-level perception-action through complex social behaviors, are associated with a general variation in learning. This suggests that a domain-general view to development, by which variation in basic processes of learning can manifest as atypical social development, should be considered as a mechanistic approach to studying autism.

## 6. Implications

Considering the role of allostasis regulation, and multi-modal perception and action in social development can impact the way autism is studied. Our framework portrays a line of new hypotheses and research questions on the neurobehavioral mechanisms of autism. Specifically, this framework calls for testing the mechanistic role of learning in autism, along with the domain-general ingredients of learning including allostasis regulation, perception (both interoception and exteroception), motor control, and multi-modal association.

This framework can also impact how autism is diagnosed and treated. As of today, the average age of diagnosis in children is three years, when developmental delays are observed in communication and social behavior [173]. The age of diagnosis is important for useful interventions that improve the child outcomes, including self-reliance and well-being [174]. Decreasing the age of diagnosis is one of the key priorities in the field of autism research [175,176]. According to our model, distinct neural and behavioral features that are potentially relevant for identifying autism can be reliably measured in very young infants. For example, current research methods enable to measure in very young infants allostasis regulation [177], parent-infant synchrony [89], the rate of learning new concepts [178,179], and the formation of multi-sensory neural tracts [180,181]. We hypothesize that variation in these measurements in early infancy can be indicative for a later prognosis for atypical social development, and can potentially serve as bio-behavioral markers for autism early on. Moreover, there is a long-standing debate regarding the clinical distinction between autism and other spectrum disorders, such as Asperger’s syndrome or pervasive developmental delay not otherwise specified (PDD-NOS) [182,183]. Future empirical research is called for in order to test whether the distinct neural and behavioral features described here, including perception, motor control, and learning can differentiate autism from typical development and other spectrum disorders in young children [182,183].

The model proposed here points to potential measures that can be taken to improve the behavioral outcome in children. For example, clinically targeting the dyad, which is considered here as the developmental medium of infants, can be operative in optimizing the regulation and development of every child. Numerous studies show the importance of family involvement in therapy programs [184,185]. Multiple intervention programs target parental care to improve the infant’s behavior, symptoms, and well-being [186,187]. For example, Naturalistic Developmental Behavioral Interventions (NDBI) [188] and Applied Behavioral Analysis (ABA) [189] are evidence-based approaches for early intervention in infants with autism. Both NDBI and ABA target parental behavior and were shown to induce significant improvements in the social development of infants [189,190,191,192]. This supports the proposed model, which emphasizes the importance of the social environment, and proposes a potential mechanism for how social regulation of allostasis determines concept learning, and therefore social learning [8]. Future research that clinically aims to improve the dyadic synchronization and allostatic regulation of children can potentially improve children’s well-being, their skill of synchrony, and potentially optimize social learning and development in both typically and atypically developing children. Moreover, our approach also points to basic processes of physiological regulation, as well as interoception as basic ingredients of optimal social development, and targets of clinical intervention that can improve children’s well-being and development.

## 7. Conclusions

We propose an alternative framework for autism, by which it is not a social disorder, but a general variation in allostasis-driven learning. This calls for both basic and clinical research to provide novel in-depth knowledge on mechanisms of autism as well as novel clinical paths for intervention that can improve the prognosis of individuals with autism and lower the age of diagnosis. This framework also invites investigating other psychopathologies with a domain-general view to better understand their mechanisms and develop new treatments.

## Figures and Tables

**Figure 1 brainsci-11-01269-f001:**
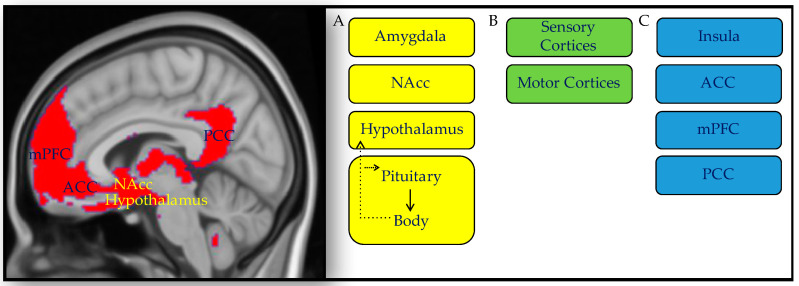
Domain-general neural circuits support social processing. Social processing in the brain does not rely on dedicated tissue, but on domain-general circuits that control allostasis, and multi-modal integration, including: (**A**) Visceromotor circuits that control allostasis, including, the amygdala, nucleus accumbens (NAcc), hypothalamus, and pituitary (in yellow) [71]; (**B**) Cortices of perception and action that perceive the environment and control behavior (in green); (**C**) Association cortices including limbic cortices in the anterior cingulate cortex (ACC) and insula, as well as the medial prefrontal cortex (mPFC) and posterior cingulate cortex (PCC) [81]. With experience, association cortices integrate multi-modal information and represent it as concepts, including emotion concepts and concepts about the mind of others [30,64] (in blue). Altered function in these circuits, which characterizes individuals with autism [82,83,84], supports the domain-general view of autism.

**Figure 2 brainsci-11-01269-f002:**
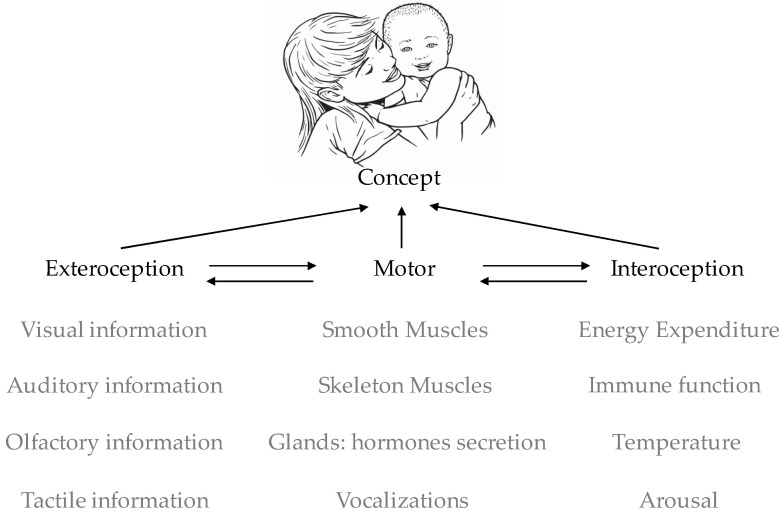
Basic ingredients of learning favor social development. During development, infants organize sensory-motor information into concepts. Information includes the exteroceptive perception of the environment, interoceptive perception of the body, execution of motor control over behavior (skeleton muscles), and visceromotor control of the internal milieu via the autonomic and endocrine systems (contraction of smooth muscles, hormonal secretion from glands). Two domain-general processes that impact learning are statistical learning of the environment [109,110,111] and motivated learning guided by allostatic demands [93]. In social animals, the environment is primarily social, and allostasis is regulated primarily via social interactions. Therefore, the first and most prominent learning humans do is social. Accordingly, a deficiency in multi-modal integration of the basic ingredients that determine the ability to learn concepts, including perception, motor control, and allostasis regulation, results in atypical social development.

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
