# Peer review of "What Is Social about Autism? The Role of Allostasis-Driven Learning"

_brainsci, 2021, doi:10.3390/brainsci11101269_

Round 1
Reviewer 1 Report
The current manuscript aimed at proposing an alternative framework for autism, by which it is not a social disorder, but a general variation in allostasis-driven learning. This calls for both basic and clinical research to provide novel in-depth knowledge on mechanisms of autism as well as novel clinical paths for intervention that can improve the prognosis of individuals with autism and lower the age of diagnosis. This is a well written manuscript and the minor concern that I have is, how easy it is to differentiate the autism from other spectrum disorders in the earlier age kids and what are the potential measures that needs to be taken to improve the outcome of behavior in children.
Author Response
- The minor concern that I have is, how easy it is to differentiate the autism from other spectrum disorders in the earlier age kids
We thank the reviewer for this interesting point. The revised manuscript now includes a discussion on additional spectrum and developmental disorders was added on page 10.
- What are the potential measures that needs to be taken to improve the outcome of behavior in children.
The revised manuscript now includes a discussion on the potential measures that can improve children’s well-being and development on page 10.
Reviewer 2 Report
The authors provide a well-written review on learning deficits in autism disorders. I have only a few minor comments.
- The title seems misleading and confusing. I suggest changing to a more straightforward title.
- Both figures are labeled as Figure 1. In the second figure, there seem to be some errors that some texts are shown up as question marks.
- While the authors focus on deficits in learning, it would be nice to discuss how social learning fits or does not fit the dopamine-mediated reinforcement learning theory, related to the VTA/RMTg.
Author Response
- The title seems misleading and confusing. I suggest changing to a more straightforward title.
The title is now revised to be more straightforward.
- Both figures are labeled as Figure 1. In the second figure, there seem to be some errors that some texts are shown up as question marks.
We corrected the figure title.
- While the authors focus on deficits in learning, it would be nice to discuss how social learning fits or does not fit the dopamine-mediated reinforcement learning theory, related to the VTA/RMTg.
We thank the reviewer for this point. The revised manuscript now includes a discussion on reinforcement learning, its underlying dopaminergic mechanism, and how they vary in autism (see Page 8-9).